# Combined Pulsed RF GD-OES and HAXPES for Quantified Depth Profiling through Coatings

**Muriel Bouttemy** [1,*] **, Solène Béchu** [1] **, Ben F. Spencer** [2] **, Pia Dally** [1,3] **, Patrick Chapon** [4] **and Arnaud Etcheberry** [1]

[1] Institut Lavoisier de Versailles (ILV), Université de Versailles Saint-Quentin-en-Yvelines, Université Paris-Saclay, CNRS, UMR 8180, 45 avenue des Etats-Unis, CEDEX, 78035 Versailles, France; solene.bechu@uvsq.fr (S.B.); pia.dally@ipvf.fr (P.D.); arnaud.etcheberry@uvsq.fr (A.E.)

[2] Sir Henry Royce Institute and the Department of Materials, School of Natural Sciences, The University of Manchester, Manchester M13 9PL, UK; ben.spencer@manchester.ac.uk

[3] IPVF, Institut Photovoltaïque d'Ile-de-France, 18 Boulevard Thomas Gobert, 91120 Palaiseau, France

[4] HORIBA Scientific, 14 Boulevard Thomas Gobert, Passage Jobin Yvon, CS 45002, 91120 Palaiseau, France; patrick.chapon@horiba.com

[*] Correspondence: muriel.bouttemy@uvsq.fr

**Abstract:** Chemical characterization at buried interfaces is a real challenge, as the physico-chemical processes operating at the interface govern the properties of many systems and devices. We have developed a methodology based on the combined use of pulsed RF GD-OES (pulsed Radio Frequency Glow Discharge Optical Emission Spectrometry) and XPS (X-ray Photoelectron Spectroscopy) to facilitate the access to deeply buried locations (taking advantage of the high profiling rate of the GD-OES) and perform an accurate chemical diagnosis using XPS directly inside the GD crater. The reliability of the chemical information is, however, influenced by a perturbed layer present at the surface of the crater, hindering traditional XPS examination due to a relatively short sampling depth. Sampling below the perturbed layer may, however, can be achieved using a higher energy excitation source with an increased sampling depth, and is enabled here by a new laboratory-based HAXPES (Hard X-ray PhotoElectron Spectroscopy) (Ga-K$\alpha$, 9.25 keV). This new approach combining HAXPES with pulsed RF GD-OES requires benchmarking and is here demonstrated and evaluated on InP. The perturbed depth is estimated and the consistency of the chemical information measured is demonstrated, offering a new route for advanced chemical depth profiling through coatings and heterostructures.

**Keywords:** pulsed RF GD-OES; XPS; HAXPES; depth profiling; crater chemistry; plasma-induced perturbation; InP; metrology; quantitative analyses





## 1. Introduction

Microelectronic or photovoltaic devices are constituted of successive layers, the structure of which is intensively studied to improve the efficiency and/or the robustness of the components. In particular, interface properties play a central role in the proper functioning of devices and impact upon their lifetime. Buried interface characterization is therefore essential to assess the physicochemical processes operating there in order to identify improved fabrication processes (e.g., modification of the layer stacks). Among the different methods to access the buried interfaces and specific areas of interest, pulsed RF GD-OES (pulsed Radio Frequency Glow Discharge Optical Emission Spectroscopy) leads to an accurate chemical diagnosis during fast depth profiling through coatings and interfaces over several tens of microns [1]. This rapid profiling capability, can therefore quickly yield to the chemical repartition when analyzing coated structures or stacks. In addition, the GD crater dimension may be varied from 2–8 mm diameter which is suitable for combination with many other chemical, optical or electronic probes that can measure directly at the crater bottom. However, the quantification from the pulsed RF GD-OES light intensities

to atomic concentrations is not straightforward when a material system involves many elements and requires a calibration step.

The calibration of pulsed RF GD-OES can be developed with a methodology combining it with XPS (X-ray Photoemission Spectroscopy) analysis in order to yield an advanced chemical characterization of objects consisting of multilayer stacks [2]. This methodology takes advantage of the two techniques: the speediness of the pulsed RF GD-OES and the possibility to stop at specific depths below the surface (using a "live view" of the depth profiles) and the chemical diagnosis (atomic composition and chemical environments) that XPS can measure inside the GD crater. The proof of concept of this coupling has already been demonstrated on a solar absorber ($CuIn_xGa_{1-x}Se_2$, shortened as CIGS) and has revealed an important point concerning the reliability of the information inside this crater [3]. We have evidenced the presence of a thin damaged surface layer resulting from plasma damages and from redeposition upon source shutdown, whose physicochemical properties differ from the one of the original layer. The determination of the nature of this overlayer and its thickness is essential for the development of adjusted procedures to remove it. We have shown that an oxidized overlayer, Ga-rich, was evidenced with the presence of metallic Ga balls induced by the heating of the plasma during profiling and redeposition at the interruption of the abrasion process. The thickness of this layer, evaluated from XPS depth profiling and from bromine chemical etching of the surface, was supposed to be about 50–80 nm.

In order to evaluate the chemical and optical perturbations inside the GD crater, we have previously studied the InP (100) semiconductor as reference compound [4]. Combining XPS, (EBSD) (Electron Backscatter Diffraction) and ellipsometry, we demonstrated that the morphology and the atomic network are more or less modified until approximatively 50 nm (resulting in partial loss of crystallinity, superficial In enrichment, and optical index modifications) but, as for CIGS, the exact impacted depth has still to be determined. On the basis of these anterior results, we explore in this work the capabilities of pulsed RF GD-OES and photoemission but with a higher energy photon source. Using a higher energy X-ray source than conventional XPS (mainly Al-K$\alpha$ radiation at 1486.6 eV), the sampling depth increased from 5 nm up to 30–50 nm, leading to a more bulk-sensitive measurement. The sampling depth is photoelectron kinetic energy dependent and may be calculated using the TPP-2M (Tanuma, Powell and Pen) formula [5,6]. For device structures with layers thinner than this maximal HAXPES (Hard X-ray PhotoElectron Spectroscopy) sampling depth, angle-resolved HAXPES may be used to increase the photoemission angle, $\theta$, and reduce the sampling depth by a factor $\cos(\theta)$, analogous to changing the photon energy at a synchrotron [7]. Additional core levels also become accessible and thus measuring different core levels in itself may provide an extended escape depth range. In the present case, HAXPES gives us the opportunity to detect the elemental concentration and chemical state within the whole damaged layer and, in many cases, probably below [8]. Experiments have been conducted on a novel laboratory-based HAXPES spectrometer using a Ga-K$\alpha$ (9.25 keV) X-ray source (ScientaOmicron GmbH) [9]. Post-mortem analysis of the GD crater realized on InP (100) substrates is presented, comparing XPS and HAXPES compositional data and spectral signatures. This study demonstrates how pulsed RF GD-OES coupled with HAXPES may be employed to fast access to deeply buried layers of complex stacks and buried associated interfaces (tens–hundreds $\mu$m deep) and then to chemically characterize below the GD damage layer with a non-destructive analysis. This methodology also addresses a larger issue concerning photoemission characterization, because samples may be measured without complex additional preparation (such as chemical pretreatments) nor further destructive Ar$^+$ sputtering inside the XPS spectrometer analysis chamber, which therefore preserves integrity of the initial buried information.

## 2. Materials and Methods

### 2.1. InP Samples

The pulsed RF GD-OES measurements and the GD craters were realized on an n-type InP (100) mono crystalline substrate (from InPACT, Moûtiers, France) presenting a S doping level of $1.6 \times 10^{18}$ and 364 μm thickness. Twin samples were prepared to perform comparative analyses on the two instruments (HAXPES and XPS). For HAXPES analyses, square pieces of $1 \times 1$ cm$^2$ around the crater were prepared to fit on standard Omicron sample plates. This size is also suitable to realize photoemission analyses inside and outside the GD crater (reference surface) on the same sample. No specific care was taken before XPS and HAXPES samples analyses. Samples were kept at an ambient atmosphere. HAXPES experiments were realized on samples aged 1 month and presenting thus a superficial oxide layer inherent to ageing. The HAXPES laboratory spectrometer is also equipped with a conventional Al source. XPS analyses were implemented in complement to the HAXPES run on the same equipment and at the same time on the XPS spectrometer on a control sample.

### 2.2. Pulsed RF GD-OES

Pulsed RF glow discharge optical emission spectroscopy (GD-PROFILER 2, HORIBA Scientific, Palaiseau, France) has been used for InP depth profiles. Operating conditions were: Ar plasma gas, anode diameter 4 mm, argon pressure 650 Pa, Applied RF power 30 W in asynchronous pulsed mode (pulse frequency 3 kHz, duty cycle 0.25). The DiP (Differential interferometry Profiling [10]) module has been used to monitor the crater depth all along the profile. A duration of 30 s sputtering was employed. The optical intensity was converted in atomic percentage with a calibration performed on similar samples by EDX (Energy-Dispersive X-ray spectroscopy; AZtec, Oxford Instruments, High Wycombe, UK).

### 2.3. XPS

XPS surface chemical analyses were carried out with a Thermo NEXSA$^{TM}$ spectrometer (Thermo Scientific$^{TM}$, Waltham, MA, USA) using a monochromatic Al-K$\alpha$ X-Ray source (1486.6 eV). The X-ray spot size was 400 μm. The Thermo Electron procedure was used to calibrate the spectrometer by using metallic Cu and Au samples internal references (Cu 2p$_{3/2}$ at 932.6 eV and Au 4f$_{7/2}$ at 84.0 eV). High energy resolution spectra were acquired using a constant analyzer energy (CAE) mode 20 eV and 0.1 eV as energy step size. Data were processed using the Thermo Fisher scientific Avantage© data system). XPS spectra were treated using a Shirley background subtraction and XPS compositions were deduced using the sensitivity factors, the transmission factor and inelastic mean-free paths from Avantage© library associated with the spectrometer. The fits were performed with Gaussian/Lorentzian mix, determined on deoxidized samples.

### 2.4. HAXPES and XPS

A novel laboratory-based HAXPES spectrometer (ScientaOmicron GmbH, Uppsala, Sweden) uses a Ga K$\alpha$ (9.25 keV) X-ray source and high electron kinetic energy analyzer [9], which has recently been benchmarked and elemental sensitivity factors calculated to enable quantification [7]. An Al K$\alpha$ X-ray source is also attached and aligned to measure the same sample position; however, the Ga X-ray spot size is microfocussed to 50 μm in order to yield sufficient flux to overcome the associated decrease in photoionization cross section at higher photon energy. HAXPES measurements were carried out in transmission mode at a grazing incidence to spread the X-rays through the surface and achieve maximum photoelectron counts. HAXPES measurements were carried out at pass energy 500 eV, with survey spectra conducted using a 1.5 mm entrance slit width (associated resolution ~2 eV) and core levels with a slit width of 0.3 mm (associated resolution ~0.5 eV). Al K$\alpha$ XPS was conducted using a 1.5 mm slit width and 200 eV pass energy for surveys and 50 eV pass energy for core levels, where the resolution for XPS is approx. 50% better than HAXPES. XPS

quantification was enabled using standard Scofield factors [11] and HAXPES using home calculated sensitivity factors using important new publications on the photoionization parameters of deeper core levels and for higher photon energies [12,13] (Table 1). Binding energy (referenced as BE along the text) scale calibration was performed using Au 4f$_{7/2}$ at 84.0 eV, and fitting (using the same approach detailed above) was performed using the software CASA-XPS©. The transmission function of the analyzer was removed using the typical kinetic energy (KE) relationship of $T(E) = KE^{-0.7}$ [14], although work is ongoing to model and measure this across all the whole KE range (0–250 eV) of this analyzer in this geometry.

**Table 1.** Sensitivity factors (Scofield factors for Al-Kα, home calculated for Ga-Kα) and photoionization cross sections concerning the photopeaks of interest.

| Photopeak | In 4d | In 4s | P 2p | P 2s | C 1s | In 3d | O 1s | P 1s | Ar 1s | In 2p$_{3/2}$ |
|---|---|---|---|---|---|---|---|---|---|---|
| Binding Energy (B. E.) (eV) | ~16 | ~122 | ~135 | ~189 | ~285 | ~444 | ~530 | ~2145 | ~3202 | ~3730 |
| Scofield sensitivity factor (Al Kα) | 2.275 | 0.742 | 1.192 | 1.18 | 1.0 | 22.54 | 2.93 | N/A | N/A | N/A |
| Home calculated sensitivity factor (Ga Kα) | 1.75 | 5.95 | 0.54 | 3.13 | 1.0 | 13.0 | 3.45 | 37.8 | 65.2 | 139 |
| Photoionization cross section (9 keV) | $7.842 \times 10^{-2}$ | $2.900 \times 10^{-1}$ | $2.681 \times 10^{-2}$ | $1.64 \times 10^{-1}$ | $5.245 \times 10^{-2}$ | $6.655 \times 10^{-1}$ | $1.863 \times 10^{-1}$ | $2.469 \times 10^{0}$ | $4.943 \times 10^{0}$ | $1.152 \times 10^{1}$ |
| Photoionization cross section (1.5 keV) | $2.645 \times 10^{-1}$ | $9.111 \times 10^{0}$ | $1.411 \times 10^{-1}$ | $1.545 \times 10^{-1}$ | $1.203 \times 10^{-1}$ | $2.790 \times 10^{-2}$ | $3.583 \times 10^{-1}$ | N/A | N/A | N/A |

## 3. Results and Discussion

### 3.1. Pulsed RF GD-OES Profiling of InP

The pulsed RF GD-OES profiles have been realized in the same conditions with identical intensities vs. time slots for In and P. Figure 1 shows the In/P intensity ratio all along a profile of 30 s.

The sputtered depth was measured during the profile giving access to the thickness vs. time slot presented in the inset, and so the sputtering time axis was converted into thickness. The etching rate is estimated to 6 μm/min as already reported [4]. The steadiness of the ratio already indicates that there is no evident preferential sputtering once the top surface has been passed.

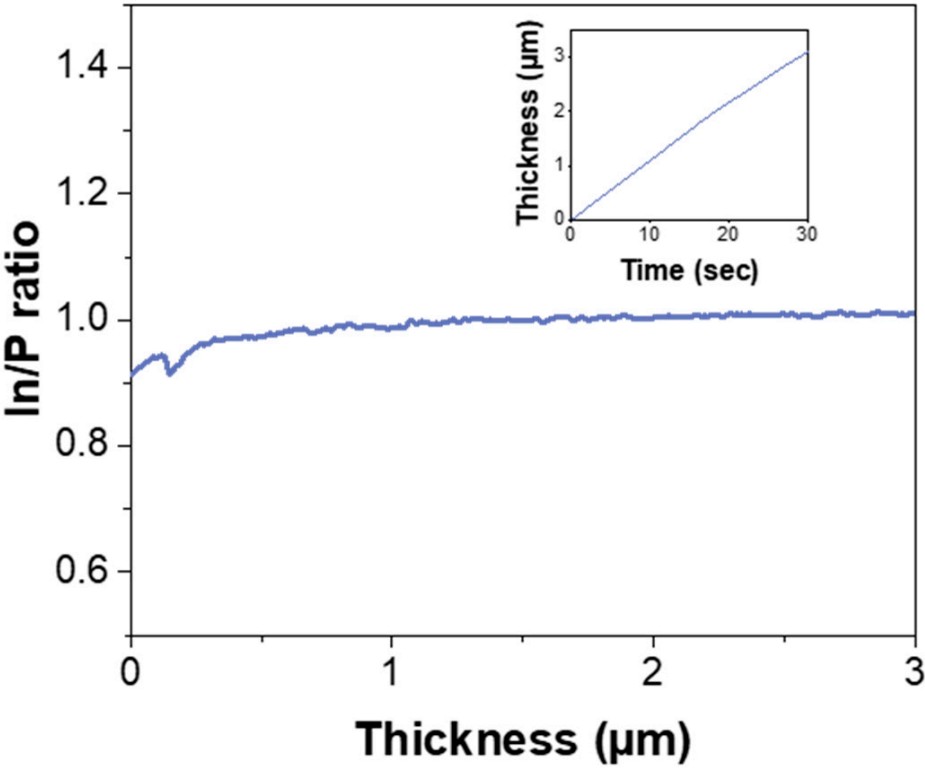

**Figure 1.** Pulsed RF GD-OES (pulsed Radio Frequency Glow Discharge Optical Emission Spectrometry) depth profile on the InP substrate. The insert corresponds to the DiP (Differential interferometry Profiling) measurement of the thickness sputtered over the plasma exposure time.

*3.2. Analysis Methodology*

As mentioned in the introduction, when interrupting the plasma profiling, the surface of the crater suffers local perturbation, then its chemical composition differs from the initial one. In a previous paper, this surface was extensively characterized using a multi-technique approach combining compositional (XPS and EDS), morphological (Secondary Electron Microscopy, SEM), microstructural (EBSD) and optical analysis (Spectroscopic Ellipsometry, SE) to get an insight into the nature of the damaged overlayer and assess the perturbed depth [4]. To complete this study, HAXPES offers the possibility to measure the bulk electronic properties and thus to probe, in a different manner than XPS, the perturbed depth inside the crater. A new generation of laboratory-based HAXPES spectrometers has already proved to be a new efficient route to characterize buried layers and bring chemical information through coatings (among them adventitious surface contamination), stretching the limits of low energy lab sources (mainly Al and [7,9]. Figure 2 shows the superimposition of the wide scans obtained on the same InP reference sample with the conventional Al-Kα source and the new generation Ga-Kα source. It clearly evidences the extended energy scale available, with the high energy source giving access to new photoelectron peaks which could not be observed at lower excitation energy, namely P 1s (BE: 2149 eV), In 2p (BE: 3938–3730 eV) and In 2s (BE: 4238 eV) in the case of InP [15,16]. The inset shows the Ga and Al X-ray photoemission spectra across the traditional XPS binding energy range up to 800 eV BE, where it can be observed that the relative intensities of the photoelectron peaks (e.g., In 3p and In 3d) change dramatically when the photon energy is increased, which is due to changes in the photoionization cross sections and therefore the sensitivity factors (as demonstrated in Table 1). These spectra also show the loss of any signal from O 1s and C 1s using HAXPES, where sensitivity to surface contamination is diminished. The sampling depth (i.e., 3 times the inelastic mean free path) can be estimated using the TPP-2M model which is carried out in the QUASES© (Quantitative Analyses

of Surfaces by Electron Spectroscopy) software [17], giving 8.9 nm for P 2p using the Al source and 34.3 nm for P 1s and 42.3 nm for P 2p using the Ga source. It is important to note that photoelectrons are still detected from the surface, as described by the Beer–Lambert law [10], but the increase in sampling depth diminishes the relative contribution from the topmost surface. To evaluate the impact of the GD-OES sputtering upon the material properties, the differences observed with different sampling depths will be investigated by choosing two sets of photopeaks for In and P, situated at low and high binding energies and within a restricted energy range to limit escape depth variation between the selected photopeaks couples. P 2p–In 3d (usually employed for conventional XPS measurements), P 2s–In 3d and P 1s–In 2p (for HAXPES measurements) are then chosen.

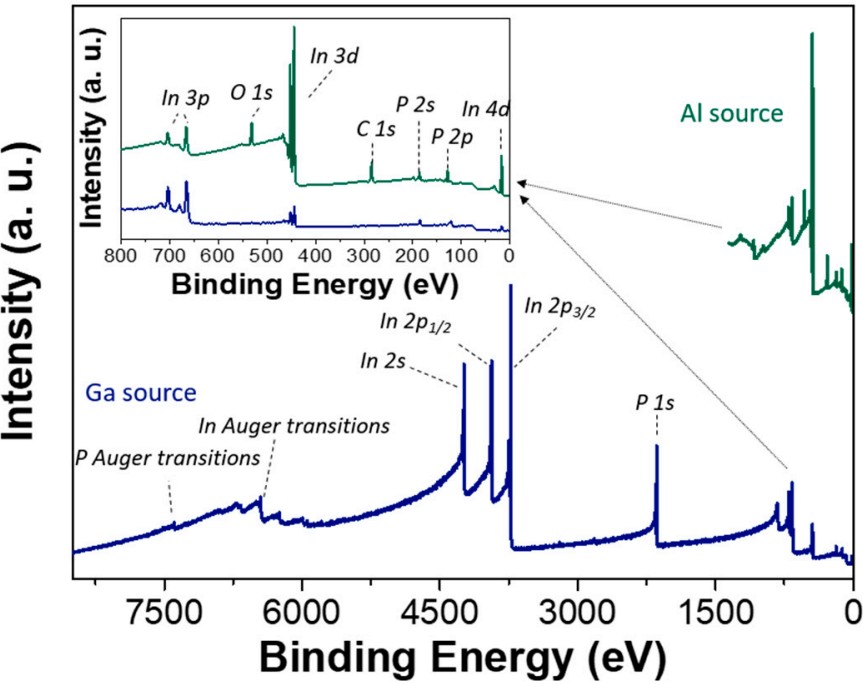

**Figure 2.** Widescans on InP substrate, using a conventional Al source (green) and a high energy Ga source (blue), with a specific inset of the 800–0 eV region. The high energy Ga source survey spectrum shows an absence of O 1s and C 1s.

### 3.3. GD Crater Characterization by Photoemission: XPS vs. HAXPES

Figure 3 presents the In 3d, P 2p and P 2s spectral regions measured outside the crater for the reference sample and inside the GD crater for the modification evaluations. The three spectral regions are recorded sequentially with the Al-Kα (XPS) and the Ga- Kα (HAXPES) sources. Outside the crater, the In 3d reference photopeak recorded with the Al-Kα source (Figure 3a) presents two contributions, related to two chemical environments. The predominant one is associated with the InP network (In $3d_{5/2}$ at 444.5 ± 0.1 eV, with a FWHM (Full Width at Half Maximum) of 0.79 ± 0.02 eV [18]) and the other to a small oxide phase (In $3d_{5/2}$ at 445.3 ± 0.1 eV, typical of In–O bounds [19], with an FWHM of 1.18 ± 0.02 eV). Inside the GD crater, obvious modifications are observed in line with a noticeable increase of the FWHM of the global envelope of the photopeak (0.81 ± 0.02 eV outside the crater compared to 1.34 ± 0.02 eV inside the GD crater). The deconvolution procedure again finds two contributions corresponding to the InP matrix and the superficial oxide environments but requires increasing FWHM (1.04 ± 0.02 eV and 1.30 ± 0.02 eV for the In–P contribution and the In–O contributions, respectively). Moreover, the contributions are slightly shifted to lower binding energies (In $3d_{5/2}$ photopeaks at 444.1 ± 0.1 eV and 445.1 ± 0.1 eV for the In-P and the In–O contributions). This enlargement, accompanied with the anomalous chemical shift, must be considered as being due to the presence of metallic indium formed inside the GD crater, already observed in similar configuration by

SEM revealing the presence of balls attributed to metallic In by complementary localized probe Auger analyses [4]. These balls can be partially oxidized after air interaction and thus contribute to the oxide contribution intensity.

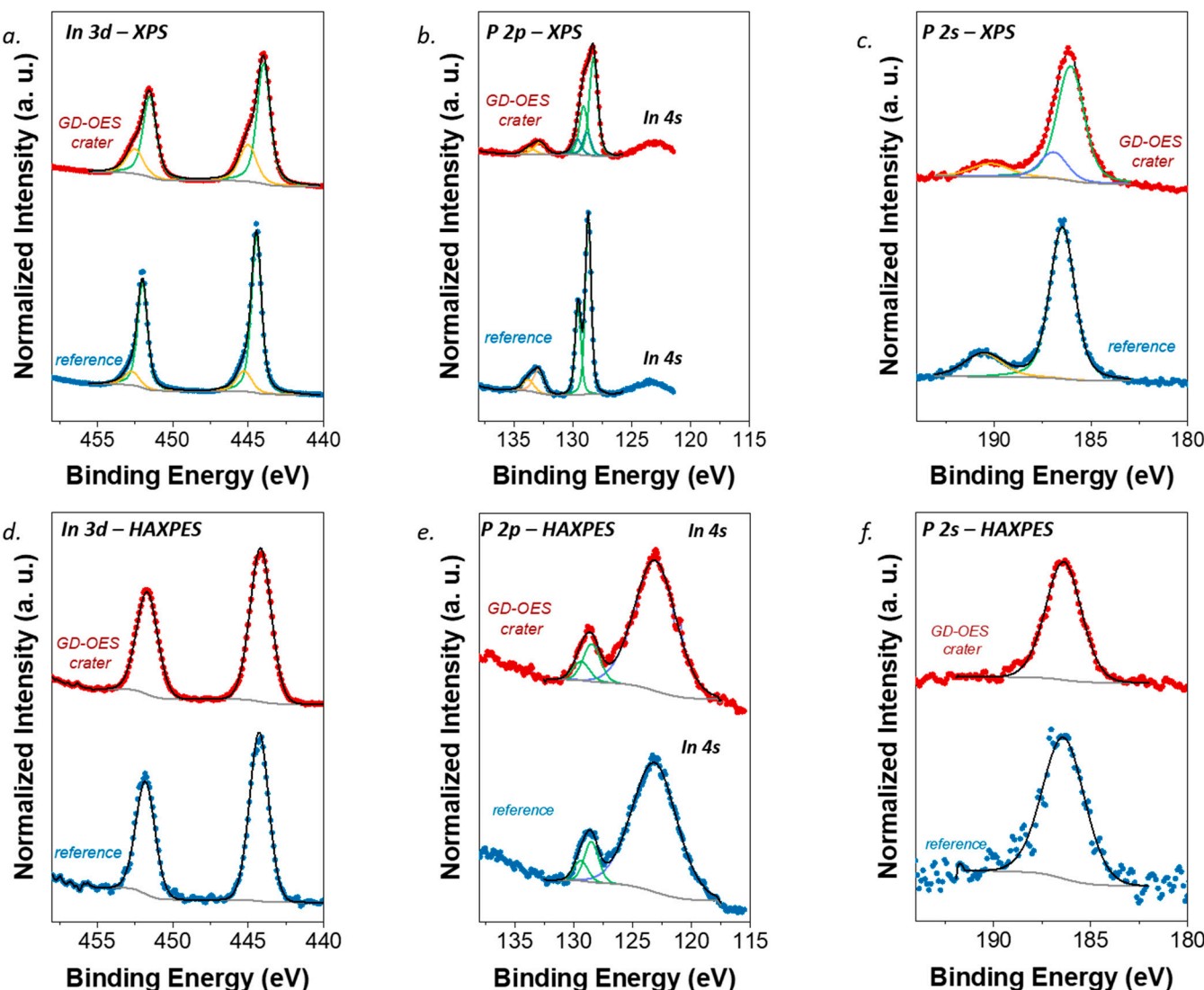

**Figure 3.** In 3d, P 2p and P 2s high energy resolution spectra for the InP surface outside (blue, bottom spectra) and inside (red, top spectra) the GD crater measured with Al-Kα (1.49 keV) (**a**–**c**) and Ga-Kα (9.25 keV) (**d**–**f**) sources. The fitting reconstruction with oxide (orange) and matrix (green) contributions is represented.

Regarding the P 2p reference spectrum (Figure 3b), two well separated chemical environments are shown outside the crater, similarly to the In 3d reference spectrum. The In–P environment (P 2p$_{3/2}$ at 128.7 ± 0.1 eV with a FWHM of 0.59 ± 0.02 eV [18]) gives rise to a well-defined spin-orbit splitting and the P–O one (P 2p$_{3/2}$ at 132.9 ± 0.1 eV with a FWHM of 1.09 ± 0.02 eV) to a well-known higher binding energy contribution. Inside the GD crater, important modifications are also noticed, with the spin orbit resolution becoming indistinct. The apparent position of the In–P contribution photopeak is also again shifted to lower binding energy and the FWHM value increases (P 2p$_{3/2}$ at 128.3 ± 0.1 eV with a FWHM of 0.83 ± 0.02 eV). Consequently, a second doublet in the low BE region of the P 2p photopeak has to be added to satisfactorily fit the spectrum (P 2p$_{3/2}$ at 128.9 ± 0.1 eV with a FWHM of 0.83 ± 0.02 eV). The oxide contribution is still present, but its position remains stable regarding the reference sample (P 2p$_{3/2}$ at 132.9 ± 0.1 eV with an FWHM of 1.21 ± 0.02 eV). Another photopeak of interest is the P 2s (Figure 3c), situated in a

close energy range (+58 eV on the BE scale) and presenting the same features as the P 2p. Outside the crater, two chemical environments (In–P at 186.5 ± 0.1 eV with a FWHM of 1.53 ± 0.02 eV and P–O at 190.6 ± 0.1 eV with a FWHM of 2.35 ± 0.02 eV) are necessary to model the photopeak (the different parameters are exposed on Table S1, SI). As expected, the energy proximity of these P 2p and P 2s photopeaks also leads to quasi identical area ratios of 0.3 between the high and low BE contributions. As for the P 2p, inside the GD crater, an additional contribution has to be considered for the P 2s low BE fitting procedure, to take account for the In–P spectral region enlargement, accompanied as well by a small shift to lower BE. These observations on In 3d, P 2p and P 2s confirm the evident perturbation of the surface atomic network, more particularly the In and P sub-networks requiring a modification in fitting parameters, and this disorganization is mainly attributed to an amorphization phenomenon [4].

As expected, using HAXPES to sample further into the surface, a different chemical diagnosis is made. The oxide contributions associated with surface oxidation are no longer visible, neither in the In 3d nor for the P 2p or the P 2s photopeaks (Figure 3d–f and fits parameters in Table S1, SI). The disappearance of the In–O and P–O contributions, concomitant with the disappearance of the O1s peak on the survey spectrum (Figure 2), is visibly in line with the expected lower surface sensitivity of the HAXPES technique. In addition, no energy shifting of the peaks is observed, as well as no FWHM broadening. By comparing In 3d, P 2p and P 2s HAXPES signals obtained outside and inside the GD crater, photopeaks are perfectly superimposed (no enlargement nor chemical shift), indicating that no significant modifications in relation with the surface perturbation inside the crater can be detected, and the surface oxide contribution appears invisible. HAXPES provides a more bulk-like picture: comparison of Figure 3b,e, presenting the P 2p–In 4s spectral region, emphasize a very interesting feature related to the strong evolution of the ionization cross section with the incident photon energy. The relative intensities of the In 4s and P 2p contributions are inverted for XPS and HAXPES, the In 4s contribution being dramatically increased compared to the P 2p photopeak with the high energy source. This highlights the importance to dispose of correct sensitivity factors for an accurate quantification [7,12,13]. Moreover, due to the peak broadening inherent to Ga-Kα excitation (due to a broader FWHM inherent to the Ga excitation), In 4s–P 2p separation becomes less evident. The In 3d–P 2s photopeak set, with isolated peaks, is here preferred to obtain quantitative data, because the photoionization cross-section of P 2s photopeak is higher than the one of P 2p (Table 1). These considerations explain why P 2s is more commonly used in HAXPES compared to P 2p.

In addition to the classical photopeak levels obtained at binding energies usually reached with an Al source, HAXPES measurements provide new photopeaks with higher binding energies, such as In $2p_{3/2}$ and P 1s positioned for InP at 3731.8 eV [15] and 2145.3 eV [16], respectively (Figure 4, fits parameters in Table S2, SI). Once again, due to the higher sampling depth, no clear modifications of In and P signatures are observed when comparing the HAXPES responses obtained outside and inside the GD crater. Concerning the In $2p_{3/2}$ level, the spectrum is easily fitted with one Gaussian–Lorentzian peak and there is no noticeable difference related to a surface state, contrary to XPS results. Regarding the P 1s photopeak, the situation is slightly different. This peak presents a small contribution at 2147.5 ± 0.1 eV whose attribution remains uncertain so far. P 1s being positioned at lower BE than In $2p_{3/2}$, it is supposed to be less sensitive to the extreme surface. One explanation could be correlated to the presence of a residual of P–O bond detection expected at this BE position [15]. The area ratio between this side signal and the main one is 0.05, which must be compared to the ratio of 0.30 obtained for P 2s or P 2p levels in conventional XPS configuration, explained by the fact that P 1s is much less surface sensitive (the kinetic energy of the photoelectrons from P 1s is ~7100 eV compared to ~1350 eV for P 2p with XPS), and therefore measures a relatively smaller amount of oxide. However, a direct attribution of this high BE contribution to P photopeaks (P 2s or P 1s) is not coherent with the oxidation process of InP, for which both In and P oxidation is expected, and

consequently, an oxidation contribution on In $2p_{3/2}$ photopeak should be visible as well. However, the In$2p_{3/2}$ peak is broader than P 1s peak, as expected given In 2p is at higher binding energy, and it could be a small oxide peak which is not detected within a peak that appears very symmetric.

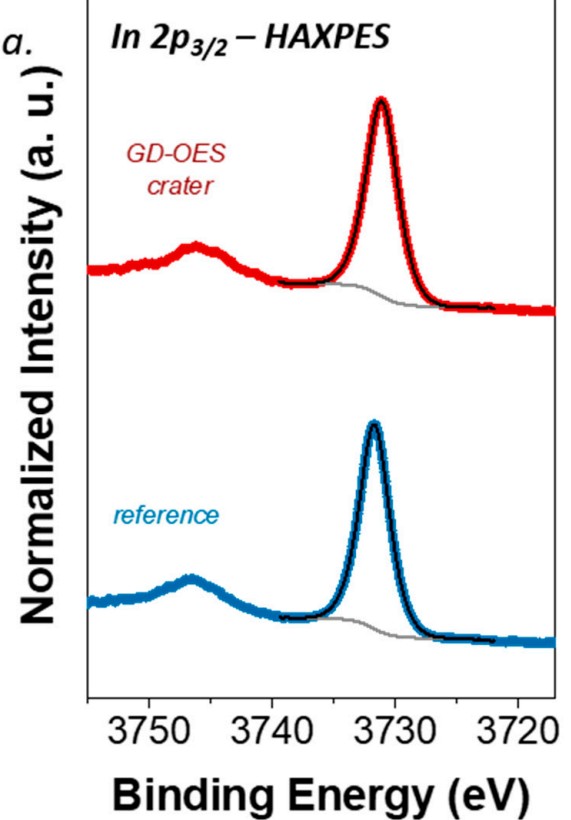 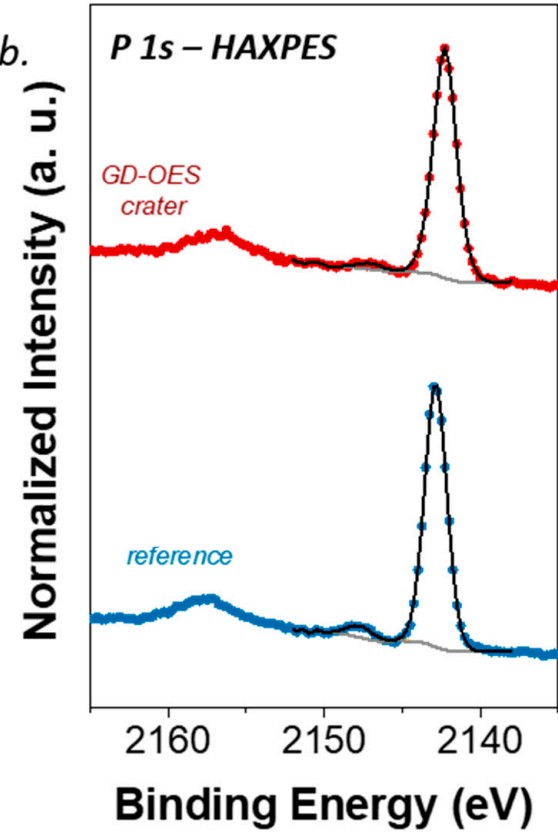

**Figure 4.** High energy resolution spectra for the InP surface outside (blue, bottom spectra) and inside (red, top spectra) the GD-OES crater, In $2p_{3/2}$ (**a**) and P 1s (**b**) with the Ga-Kα (9.25 keV) source.

Quantitatively speaking, the XPS In/P value of 1.29 reveals (Table 2), as expected, an In enrichment at the GD crater surface, in agreement with the presence of metallic In formed when the dynamic profiling of GD is stopped. However, when comparing the measurements performed in HAXPES, the ratio In/P goes back to a value very close to the nominal 1, both outside and inside the GD crater analysis. These results therefore demonstrate that a stoichiometric ratio is measured using HAXPES, due to its deeper sampling depth of ~ 30 nm. These measurements allow us to assess that the GD damage layer is of the order of 10 nm or less, given the surface damage measured using a traditional Al source (<10 nm depth probed).

**Table 2.** In/P atomic ratios determined using XPS P 2s–In 3d and HAXPES P 1s–In $2p_{3/2}$ set of photopeaks.

| | In/P | | | | | |
| --- | --- | --- | --- | --- | --- | --- |
| | P 2s–In 3d (XPS) | | | P 1s–In $2p_{3/2}$ (HAXPES) | | |
| – | Global | Matrix | Oxide | Global | Matrix | Oxide |
| InP ref | 1.01 | 1.04 | 0.88 | 1.05 | 1.05 | 0.00 |
| InP GD-OES crater | 1.29 | 1.06 | 2.31 | 1.08 | 1.08 | 0.00 |

## 4. Conclusions

The composition inside a GD crater has been measured by means of photoemission using two excitation sources: Al Kα emitting at 1.49 keV (XPS) and Ga Kα 9.25 keV (Lab HAXPES). The use of a higher energy source enables chemical state analysis to reach a higher sampling depth (of the order of three times greater here) and to access core level photopeaks that were not accessible with a conventional XPS spectrometer. The combined use of XPS and HAXPES enables a complete investigation of the composition inside the crater, knowing that previous results have already demonstrated the presence of a perturbed overlayer inherent to the GD profiling interruption. Comparing XPS and HAXPES results allows the corresponding perturbed depth to be evaluated of the order of 10 nm, refining the previous estimation determined from spectroscopic ellipsometry and EBSD measurements. HAXPES presents the advantage to be less surface sensitive, bringing accurate "bulk" chemical information (up to 50 nm sampling depth and more when considering elastic and inelastic background interpretation [7]). This represents real added value to perform chemical characterization through coatings and at buried interfaces. The combined use of GD-OES and XPS can therefore efficiently access deeply buried locations, where the reliability of traditional XPS analysis inside the crater is limited by the presence of the perturbed layer, requiring an intermediate step to regenerate the surface. This step is not required when using HAXPES, which can effectively neglect the surface degradation, and this therefore broadens the areas of application for this technique, particularly when measuring buried interfaces and depth profiling through coatings. The ability to verify the composition below the surface perturbation layer therefore enables the straightforward calibration of GD-OES depth profile voltage signals, which is likely to lead to a wider uptake of GD-OES for complex, multi-element materials that require quantification of atomic concentrations; quantification through stacked layer structures may easily be performed by using GD-OES to depth profile through each layer, and measure with HAXPES as each interface is removed. Additional information about chemical environments can also be obtained. Further work is planned to verify this protocol, as applied to a variety of materials, but this proof of principle study demonstrates the potential for combined GD-OES and HAXPES for a wide range of advanced materials research.

**Supplementary Materials:** The following are available online at https://www.mdpi.com/article/10.3390/coatings11060702/s1, Table S1: Evolution of the position and FWHM parameters for In $3d_{5/2}$, P $2p_{3/2}$ and P 2s for XPS and HAXPES measurements, Table S2: Evolution of the position and FWHM parameters for In $2p_{3/2}$ and P 1s for HAXPES measurements.

**Author Contributions:** M.B., Conceptualization, methodology, validation, formal analysis, supervision, writing—original draft preparation, writing—review and editing; S.B., methodology, validation, formal analysis, writing—original draft preparation, writing—review and editing; B.F.S., Conceptualization, methodology, validation, formal analysis, supervision, writing—original draft preparation, writing—review and editing; P.D., formal analysis. P.C., Conceptualization, methodology, writing—original draft preparation; A.E., methodology validation, supervision, writing—original draft preparation, writing—review and editing. All authors have read and agreed to the published version of the manuscript.

**Funding:** This work was supported by the Henry Royce Institute, funded through EPSRC grants EP/R00661X/1, EP/P025021/1 and EP/P025498/1.

**Institutional Review Board Statement:** Not applicable.

**Informed Consent Statement:** Not applicable.

**Data Availability Statement:** Data available in a publicly accessible repository. The data presented in this study are openly available in Figshare with DOI: https://doi.org/10.48420/14754519.

**Conflicts of Interest:** The authors declare no conflict of interest.

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
