# Peer review of "Combined Pulsed RF GD-OES and HAXPES for Quantified Depth Profiling through Coatings"

_coatings, doi:10.3390/coatings11060702_

Round 1

Reviewer 1 Report

The manuscript deals with chemical characterization at buried interfaces and presents a methodology combining pulsed RF GD-OES and XPS. The analysis is affected by a perturbed layer present at the crater surface. The approach proposed by the authors combines HAXPES with pulsed RF GD-OES and represents a new route for advanced chemical depth profiling through coatings and heterostructures. The manuscript is original, quite interesting, well written thus deserves to be published in the present form.

Reviewer 2 Report

This article concerns the combination of pulsed 35 RF GD-OES (Glow Discharge Optical Emission Spectroscopy) and Hard X-ray Photoelectron Spectroscopy (HAXPES) to evaluate depth profile which allows verifying the composition inside crater. X-Ray photoelectron spectroscopy (XPS) and HAXPES results are compared which conclude that HAXPES presents the advantage to be less surface sensitive, 323 bringing accurate “bulk” chemical information (up to 50 nm sampling depth and more 324 when considering elastic and inelastic background interpretation). The advantage of the method is to enable accessing buried locations with the presence of a perturbed layer.

The article is interesting, and the general structure of the article is well presented. However, it presents an exploratory research result. In order to be able to draw an exhaustive conclusion, the method needs to follow a rigorous design of experiment to ensure the reliability of the results. Besides, the article presents no future work perspectives which give me an impression as the article is just presenting the results of some experimentation without looking further into its utilization in industrial or academic application. Moreover, the quality of presentation can be improved. The article suffers the rupture of concepts and it is presented as blocs of concepts and results. It can be improved by valorizing the article using chained tests and descriptions. The quality of figures, such as figure 2, can also be improved by adding descriptive information.

Reviewer 3 Report

Bouttemy and al. present the results obtained by testing HAXPES coupled to RF GD-OES as a possible technique for depth profiling analysis. RF GD-OES is a rapid technique for the depth profiling analysis but it requires a calibration step. The coupling of RF GD-OES with XPS or HAXPES permits a rapid depth profiling analysis thanks to the fast etching rate of the RF GD and a good sample analysis thanks to the XPS or HAXPES surface analysis. The authors compare with classic XPS analysis the results obtained by HAXPES both to demonstrate the effect of RF GD-OES on the sample surface and to evidence the better results obtained by HAXPES.

Essentially the authors demonstrate that the RF GD-OES modifies the surface of the sample and the analysis made with XPS can be affected by errors due to the presence of oxides and redeposited materials after etching. The higher HAXPES sampling depth permits obtaining correct results by analyzing a higher section of the sample.

The article is well written and clear and the topic is interesting. In my opinion the article can be published after very minor revisions.

I have just a critical point on the technique that probably can be addressed in the article. Authors presented the technique for the analysis of buried interfaces and they proposed HAXPES as analysis method. The high sampling depth of this technique permits to overcome the problem of the modified surface of the sample but makes it a bulk technique and no longer a surface technique (as instead is XPS). This fact can be a problem for the analysis of very thin films and probably it cannot be used to obtain information about the interface.

Author Response

Plese see attachment.

Reviewer 4 Report

This paper reports about a modification of a method that combines XPS and GD-OES to obtain detailed chemical information in depth. The idea is to combine the speed of GD-OES and the detailed output from XPS (mainly chemical bonding, since although composition is also mentioned, it is already obtained by GD-OES). That method was proven in several previous papers, and in the present one they examine the advantage of using Hard-rays XPS (Ga source), labeled as HAXPES, instead conventional XPS in the crater. The paper is clear and well organized, although I have some important concerns about it. Therefore, I recommend the following major revision.

The main conclusion that I obtain from the paper is that, when using HAXPES, a larger depth is probed (50 nm) than when using XPS. As a result, there is no difference between the results in the depth of the crater and the surface, contrary to what is observed in conventional XPS. Such difference is attributed to the influence of the crater erosion. In other words, the surface of the crater is modified to what ‘in theory’ is.  I have two problems with that:

The first one is that the suggested solution is to use a technique with lower depth resolution (HAXPES). However, the main purpose of the overall method is to increase the resolution of GDOES. Therefore, they solve a ‘problem’ with a ‘solution’ that goes against the overall purpose of the technique. For instance, for thin layers (e.g. 5 nm), HAXPES would be useless.

The second one is about the results themselves. The differences between curves in Figures 3a, b and c is not that large to consider that it is necessarily caused by the ‘crater effect’. To prove that, more samples are needed (one type of sample is not enough for such statement), since other effects related to that sample itself could be responsible of that (maybe the sample is different in the surface than in the bulk; similar effects are observed after a conventional Ar cleaning). In addition, the data analysis needs revision. For instance, the main contribution of P2s is not located at the same energy in both XPS plots. In fact, I cannot understand why the main P2s peak in the crater is not centered in the maximum. In addition, the chemical shift of the peaks can be caused by a different reason (e.g. the crater effect causes a difference in charging, the external surface is actually different from the surface of the crater, due to e.g. different conditions of exposure to ambient). It is in fact curious that all the contributions shift in the same direction (did the authors calculate the Auger parameter?). Furthermore, the explanation for the new contributions (amorphization) seems quite speculative. XPS is basically sensitive to the ‘near environment’, and I would like to see which ‘amorphous phase’ different from the other phases in the sample is behind those peaks, and which other contributions appear in other XPS peaks (i.e. the new binding energies have to be assigned properly to a new phase in all the XPS peaks, not only in the P peaks: what phase are these contributions connected to?).

Regarding the ‘benefit’ of HAXPES, it is true that the spectra look the same inside and outside the crater. However, the width of the peaks is much bigger! Therefore, maybe we are seeing the same due to the lack of energy resolution. In other words, if we apply an ‘adjacent averaging’ on the conventional XPS spectra, we may arrive to similar results as well. In other words, maybe the ‘same result’ is just a consequence of lack of lateral resolution, and the ‘problem’ is just good energy resolution.

Other comments on the paper:

  • Line 165: the etching rate is set to 10 microns per minute, but in the inset of Fig. 1 the thickness is 3 microns in 30 seconds (not 5). Please, correct.
  • Lines 216-220. Here the authors claim about the presence of metallic indium. However, I failed to see any peak of that phase. That phase has to appear, particularly if they are ‘balls’ as described in the text.
  • Lines 201-245 (and any other where applicable): Include a table with the position, FWHM with errors of all the contributions. Reading all that data in the paper, without having them summarized in a Table is unnecessarily complicated.
  • The XPS figures need to be properly plotted. Try to adjust the X axis only to the ‘real data’. For instance, plot only from 132 eV in Fig 3b, and e. In Fig.3e, make sure that the end of the fitting is seen (ca. 110 eV), and use the same scale in b. In a and d adjust as well (441 to 455) and the same in c and f. Adjust the x axes in such a way that that there are numbers at the beginning and at the end of each axis. Furthermore, make sure that the fitting line (black) only fits the area where the Shirley background is displayed (region of fitting), but it is not included in the region where only experimental data is plotted.
  • Line 254-255: Comment also that the O1s contribution disappears in the overall spectrum (Fig. 2).

Reviewer 5 Report

The authors represents valuable results from the ongoing development of HAXPES method which allows to obtain chemical information from the layers which are not detectable by the conventional XPS with Al or Mg radiation. The possibility to look beyond the layer which is altered by plasma etching inside GD-OES or Ar+ sputtering inside the XPS spectrometer might be very useful for various fields like solar cells where layered structures are crucial for material functionality. The manuscript is well written and clearly presents main findings and directions of the research.

Interestingly, qualitatively the presented data leaves an impression that the the fraction of photoelectrons detected from the top surface by HAXPES is relatively small to what one would expect. Therefore, looking forward to the new results of the method development it would be interesting to get more details on the distribution photoelectron generation and acquisition from the different thicknesses of the sample. For example, in addition to the presented analysis of the altered InP layer inside GD-OES created crater it would be also interesting to see how this HAXPES/XPS combination would apply to the analysis of unaltered, intentionally deposited over-layers with definite thicknesses and compositions.

Looking for the possible improvements of the article I would suggest to use the same binding energy scale for Fig. 3b and Fig. 3e (also to correct letter b to letter d in the Fig. 3 caption). 

Round 2

Reviewer 4 Report

As a reviewer, I am not convinced with the content of the manuscript nor with the responses of the authors. Nevertheless, I do not want to delay the production of the paper with circular discussions, and therefore I leave the final decision to the Editor. It is probably me who fails to understand the point of the paper.

First of all, as indicated by the authors in the responses to the reviewers and the manuscript, this paper belongs to a series of papers. It some points is difficult to ‘decouple’ the present manuscript with the results of the other works, which I had no time to look. In that regard, it is complicated to judge the degree of precision of some sentences and observations, since the authors refer to those previous results that I did not check (e.g, the presence of metallic elements in spheres measured by SEM; are we talking about the same exact specimen or not?).

Regarding the content of the paper, it is true that the HAXPES avoids the differences between the surface of the sample and the surface of the crater. However, this comes at expenses of losing depth resolution (the new spectra averages information from 50 nm), and losing lateral resolution (i.e. resolution in the x axis, binding energy). Both problems are not properly discussed in the paper (in fact, the authors replied in the responses to the reviewers that they will not comment on that). Regarding the first issue, they replied that ‘we do not aim to “increase the resolution” as such’. However, the introduction starts with the following sentence: ‘Microelectronic or photovoltaic devices are constituted of successive layers, the structure of which is intensively studied to improve the efficiency and/or the robustness of the components’. Those structures are normally formed by a sequence of layers whose thickness is below those 50 nm.

The second concern is of ultimate importance; XPS gives two types of information: chemical composition (which should be redundant with the GD-OES profiles), and chemical bonding. I suppose that, although the authors replied to me that ‘the aim of this methodology is to verify the material composition below the perturbation layer within the GD-OES crater, for a precise calibration of GD-OES photovoltage signal to atomic concentrations’, we are not talking here about the chemical composition, since i) the GD-OES can be calibrated, and ii) there would be no need of fitting the XPS spectra.

Therefore, to evaluate chemical bonding, you need the lateral resolution indicated before (to perform fittings and so on), and the new spectra is clearly broader than the previous (cf. FWHM data). In fact, I seriously doubt that with those broadenings you would appreciate the differences between both surfaces observed by conventional XPS. Therefore, you may be missing chemical details that are relevant. I failed to find any fruitful discussion on these issues, nor give a valid pros/cons evaluation of each approach.

In addition, some of the results are still unclear. As I mentioned before, we are referring only to one sample, which is ‘too limited’ to benchmark a new technique. In addition, the supposed ‘metallic’ contribution, which would agree with the spheres observed by SEM, is missing. The authors failed to explain how is that possible. Basically, they claimed that ‘it was too weak’, which is hard to believe if In spheres were observed by SEM, or that ‘it was too close to other contribution’ (0.3 eV), which should be distinguishable considering the errors in the position of the peaks (0.1 eV). Furthermore, no comments were done regarding some other points (e.g. the measurement of the Auger parameters).

In short, I do not have a clear picture of what do want the authors to demonstrate with the paper. If it is just that HAXPES measures the same ‘in and out the crater’ in order to make a calibration of the chemical composition of a GD-OES, it is clearly demonstrated (although I do not really see what for). However, I believe that they want to go further, and there the situation gets more complicated.
